# Self-Retained, Sutureless Amniotic Membrane Transplantation for the Management of Ocular Surface Diseases

**DOI:** 10.3390/jcm12196222

**Published:** 2023-09-27

**Authors:** Hsun-I Chiu, Chieh-Chih Tsai

**Affiliations:** 1Department of Ophthalmology, Taipei Veterans General Hospital, Taipei 112, Taiwan; hsunichiu@gmail.com; 2School of Medicine, National Yang Ming Chiao Tung University, Taipei 112, Taiwan

**Keywords:** amniotic membrane transplantation, ProKera^®^, cryopreserved, cornea surface disease, non-ophthalmologist

## Abstract

Amniotic membrane (AM) has anti-inflammation, anti-fibrotic, and regenerative effects. Sutureless cryopreserved AM transplantation, ProKera^®^ (Bio-Tissue, Inc., Miami, FL, USA), is easily applied by ophthalmologists in the treatment of ocular surface diseases. This retrospective study included patients with ocular surface diseases who received ProKera^®^ between January 2022 and May 2023. Six patients (9 eyes) with a mean age of 56.8 ± 20.8 years old (range 25–74) and a mean follow-up period of 7.8 ± 4.1 months (range 1–12) were included, including 2 of recurrent conjunctival tumors with limbal and corneal involvement (cases 1–2), 1 of pterygium with marked astigmatism (case 3) and 3 of Stevens–Johnson syndrome (SJS, cases 4–6). ProKera^®^ was inserted after the lesion excision and deep keratectomy in cases 1–3, and no recurrence or corneal complication was noted. Cases 4–5 were discharged from the intensive care unit and presented with severe chronic SJS. Most ocular manifestations improved significantly after symblepharon release and ProKera^®^ insertion, except for corneal conjunctivalization in 1 eye (case 5). Case 6 involved early ProKera^®^ use at the bedside during acute SJS, resulting in complete resolution. We concluded that the adjunctive application of ProKera^®^ can be effective for ocular surface reconstruction and provides options to intervene earlier for outpatients or patients unstable for invasive surgical intervention.

## 1. Introduction

Amnion membrane (AM) was first introduced in ophthalmology by de Rotth in 1940 [1] and a recent multicentered study demonstrated AM transplantation improved the visual acuity in patients with non-healing corneal ulcers [2]. AM transplant is widely used for various ocular surface conditions, including ocular surface reconstruction, post-pterygium excision, limbal stem cell deficiency, persistent epithelial defects, corneal ulcers and ocular surface burns [3].

AM has regenerative and anti-inflammation or anti-fibrotic effects. Koizumi et al. [4] demonstrated that AM contains many growth factors, such as epithelial growth factor (EGF), hepatocyte growth factor (HGF) and keratinocyte growth factor (KGF), which can enhance the adhesion of basal epithelial cells. Transplanting AM can provide benefits for wound healing by suppressing interleukin alpha and interleukin 1 beta in epithelial cells and acting as a barrier to inhibiting apoptosis [5]. Additionally, AM can reduce the expression of transforming growth factor-beta (TGF-beta) in fibroblasts, which may have an anti-fibrotic effect [6].

The conventional method of attaching an amniotic membrane (AM) to the ocular surface is through the use of sutures. AM is harvested from the human placenta during a caesarean section and preserved until it is needed for surgical use. Although cryopreserved AM has lower concentrations of proteins and growth factors, it retains histological properties similar to fresh AM [4,7,8]. ProKera^®^ (Bio-Tissue, Inc., Miami, FL, USA) is a self-retaining, sutureless, cryopreserved amniotic membrane that was approved by the FDA in 2003 and introduced to our country in 2020. There are two sizes of ProKera^®^ available at our hospital. ProKera^®^-classic consists of a dual symblepharon ring system, which helps to maintain orbital space and prevent adhesions. ProKera^®^-slim has a reduced height and slimmer design (0.7 mm compared to 1.1 mm for ProKera^®^-classic), which allows for better contouring to the ocular surface and improved comfort.

In this study, we presented 6 cases with the use of self-retained, cryopreserved sutureless AM, including 2 cases of recurrent conjunctival corneal tumors, 1 case of pterygium with marked corneal involvement and 3 cases of Stevens–Johnson syndrome (SJS).

## 2. Methods

This study was conducted according to the Declaration of Helsinki and approved by the Institutional Review Board for Human Research of Taipei Veterans General Hospital, a tertiary hospital in Taipei, Taiwan (IRB 2023-04-001AC). From January 2022 to May 2023, 6 patients (9 eyes) who received self-retained cryopreserved sutureless AM (ProKera^®^, Bio-Tissue, Inc., Miami, FL, USA) insertions by a single surgeon (CC Tsai) were retrospectively enrolled in the study. ProKera^®^ insertion were under topical anesthesia and operations were under topical and/or local anesthesia. During each follow-up examination, slit lamp, fluorescein staining, photographs of the ocular surface, meibomian gland images and histopathologic results were carefully reviewed. The outcomes were analyzed until the recent follow-up. 

### Amniotic Membrane Transplantation

ProKera^®^ (classic or slim, Bio-Tissue, Inc., Miami, FL, USA) was initially in a frozen form and thawed at room temperature. After rinsing with saline, ProKera^®^ was inserted under topical anesthesia with 0.5% proparacaine hydrochloride eye drops. We inserted it to the upper fornix while the patients looked down and then slid it to the lower fornix. ProKera^®^ was replaced with a new piece when melting or with heavy accumulation of debris. Postoperatively, topical non-preservative artificial tears were applied 6–8 times per day and were tapered off gradually. Topical steroids 4 times per day (prednisolone acetate 1% or fluorometholone 0.02%, depending on the disease severity) and prophylaxis chloramphenicol 0.25% 4 times per day were prescribed for 1 week and then tapered off gradually. 

## 3. Results

A total of 6 patients (9 eyes) were included in our study. The mean age was 56.8 ± 20.8 years old (range 25–74 years old). The mean follow-up period was 7.8 ± 4.1 months (range 1–12 months). ProKera^®^ was inserted initially during an outpatient operating room (case 1–5) and then exchanged at outpatient clinic (case 4 and 5). Case 6 received ProKera^®^ insertion at the bedside. Following this, we demonstrated 6 cases with sutureless cryopreserved amniotic membrane transplantation, including 2 of recurrent conjunctival tumors with limbal and corneal involvement, 1 of pterygium and 3 of Stevens–Johnson syndrome. Clinical demographics and treatment outcomes are summarized in Table 1.

### 3.1. Recurrent Pigmented Conjunctival Tumors

Case 1: A 25-year-old female came to our clinic presenting with a recurrent pigmented conjunctival lesion in the right eye. She had received total resection of a large pigmented conjunctival mass with the diagnosis of inflamed juvenile conjunctival nevus in the same location 4 years ago. Examination revealed a recurrent pigmented conjunctival lesion with corneal invasion in the right eye. There were no other ocular abnormalities. She underwent a deep tumor resection into the conjunctival and limbal regions, keratectomy for the corneal lesion, conjunctival wound suture and insertion of ProKera^®^-classic. ProKera^®^ was removed 11 days later and ideal healing was found in the conjunctiva and cornea. A stable surface with faint corneal haziness was found 12 months later. The new pathology was compound nevus.

Case 2: A 36-year-old male presented recurrent pigmented conjunctival lesions in the left eye. He received total resection of a large pigment conjunctival mass with the diagnosis of primary acquired melanosis (PAM) with moderate atypia at the same location 19 months ago (Figure 1A) and laser-assisted in situ keratomileusis (LASIK) surgery 16 years ago. Examination revealed a recurrent pigmented conjunctival lesion with corneal and LASIK flap invasion in the left eye (Figure 1B). There were no other ocular abnormalities. Delicate tumor excision, keratectomy for the corneal lesion and ProKera^®^-Slim insertion were performed. Conjunctival, limbal and corneal defects healed in 3 days (Figure 1C,D), and ProKera^®^ (dissolved partially) was removed in 5 days. No epithelial downgrowth or other flap complications were noted 1 month after operation (Figure 1E). After 7 months, there was superficial peripheral vascularization, without tumor recurrence (Figure 1F). Intact LASIK flap and no tumor recurrence were found during a 11-month follow-up period. The new pathology diagnosis was PAM with mild atypia.

### 3.2. Pterygium

Case 3: A 69-year-old female complained of blurred vision and irritation in her right eye. Best corrected visual acuity (BCVA) was 6/30 (refraction error due to high astigmatism) in the right eye and 6/7.5 (+1.75, −0.50 × 40) in the left eye. Examination revealed a giant pterygium with corneal involvement in both eyes (more severe in the right eye) (Figure 2A). There were no other ocular abnormalities except mild cataracts in both eyes. She underwent pterygium excision in the right eye, following keratectomy for the corneal lesion, conjunctival suture with 2 mm bare sclera, and the insertion of ProKera^®^-slim (Figure 2B). Conjunctival and corneal defects healed completely in 3 days (Figure 2C,D) and ProKera^®^ was removed in 6 days (Figure 2E,F). BCVA improved to 6/6.7 with minimal astigmatism (+2.00, −0.50 × 144 OD) one-day post-operation. No pterygium recurrence occurred, and BCVA remained 6/6.7 one month after operation.

### 3.3. Stevens–Johnson Syndrome

Case 4: A 74-year-old man had an intrahepatic cholangiocarcinoma under oral ivosidenib. He developed acute Stevens–Johnson syndrome (SJS) with ocular involvement after 2 days of use of oral cefixime for pneumonia. He presented with reddish papules and plaques on the trunk and 4 limbs and erosive lesions over the oral mucosa and genital area. Slit lamp examination revealed severe inflammation and ulceration in both eyelids, bilateral sloughing of bulbar and palpebral conjunctivae, and superficial punctate keratitis and filament formation on the cornea. Artificial tears and topical steroids were prescribed by internal physicians. However, after 2 months of eyedrop use, he complained of progressive vision decline with BCVA 6/30 in both eyes. Slit lamp examination showed marked symblepharon in the conjunctiva and corneal epithelial defect in both eyes. Symblepharon was released and ProKera^®^-classic was inserted under topical anesthesia. Ten months post-SJS onset and 8 months post-procedure, BCVA remained stable (6/15 in both eyes), except for little symblepharon in the peripheral conjunctiva and superficial punctate keratitis. Meibomian gland disorder and dry eye occurred; thus, punctal plugs were inserted into both eyes. 

Case 5: A 69-year-old woman suffered from facial edema with severe edematous change of lips, bilateral red eyes, and some scattered vesicles on the limbs and posterior neck after 2-day use of unknown medication for common cold, compatible with the diagnosis of acute SJS. She was sent to the emergency room due to respiratory failure and an endotracheal tube with mechanical ventilation was performed. Later, she was admitted to the intensive care unit and steroid eye drops were prescribed. As recorded, the BCVA was 6/20 in the right eye and 6/15 in the left eye. Two months after symptom onset, she presented to our clinic with tracheostomy. BCVA was 6/12 in the right eye and counting fingers in the left eye. Prominent symblepharon, discharge, severe congested conjunctiva in both eyes, and severe inflammation over the limbal area in the left eye were found (Figure 3A). Topical steroids, lubricants, and adjunctive with ProKera^®^-classic were prescribed. Meibomian gland photographs revealed severe shortening in the lower part and loss in the upper part (Figure 3C). Following 4 months of SJS onset and 2 months of ProKera^®^ use, corneal conjunctivalization with neovascular formation in the left eye developed (Figure 3B). Vision could be maintained in the right eye but was still poor in the left eye (6/6 OD and 3/60 OS) at a 7-month follow-up.

Case 6: A 68-year-old woman presented with bilateral red eyes and prominent mucosal erosions involving the oral cavity and genital area after 13 days of oral carbamazepine use for anxiety disorder. Human leukocyte antigen typing revealed a B-1502 positive. Upon arrival to the emergency room, BCVA revealed 6/7.5 in the right eye and 6/10 in the left eye. The ocular exam showed swelling conjunctiva and eyelid, as well as superficial punctate keratitis of the cornea in both eyes (Figure 4A). Following 9 days of acute SJS onset, there was prominent discharge and pseudomembrane formation in both eyes (Figure 4B) and severe oral mucosal erosions (Figure 4C). ProKera-classic^®^ was applied to both eyes on the same day at the bedside, in addition to non-preservative artificial tears (Optive fusion) every 2 hours and 1% prednisolone suspension 4 times per day. After 9 days of use, conjunctival inflammation and symblepharon subsided with a completely healed corneal defect, and ProKera^®^ of both eyes were removed. Following 3 months of SJS onset and ProKera^®^ use, there was no recurrence of conjunctival inflammation, cornea defect, or symblepharon formation (Figure 4D). BCVA achieved 6/6 in both eyes and was maintained well at a 6-month follow-up.

## 4. Discussion

This study presented six cases of sutureless cryopreserved amniotic membrane treatment in ocular surface diseases. Cases 1 and 2 had recurrent conjunctival tumors with limbal and corneal involvement. ProKera^®^-slim was used after tumor resection, and there was no tumor recurrence within follow-up. Case 3 had pterygium with marked corneal involvement. Excision combined with ProKera^®^-slim use, instead of the sutured conjunctival autograft or antifibrotic agent application, preserved most normal conjunctiva tissue and reduced the risk of scleral thinning. Cases 4 and 5 presented with chronic Stevens–Johnson syndrome. ProKera^®^-classic improved the persistent corneal epithelial defect, maintained the orbital space, and reduced symblepharon and inflammation. Visual acuity may improve, while sequela of chronic SJS influence prognosis. Case 6 demonstrated ProKera^®^-classic use in a patient with acute SJS. Early use of AM at the bedside, especially for patients with critical systemic conditions, preserves a better vision prognosis.

In cases of recurrent conjunctival tumors with corneal involvement, deep resection could reduce tumor recurrence. However, limbus contains stem cells, and damage to limbus causes the risk of wound healing and proliferation. Alternative methods include anti-cancer drugs, such as mitomycin C, interferon-alpha or 5-FU, applied adjunctive intraoperatively or topically during the postoperative period; however, those medications can be potentially toxic and cause the complication of cornea melting, scleral necrosis and sclerokeratitis. Double freeze–thaw cryotherapy is another option to reduce residual malignant cells at the surgical margins, but it may affect wound healing in the limbal area. In case 1, anti-neoplastic agents were not considered due to the benign nature of conjunctival nevus. Double freeze–thaw cryotherapy was not favored, as it can harm limbal stem cells. In case 2, despite primary acquired melanosis with atypia being a premalignant lesion, antineoplastic agents or double freeze–thaw cryotherapy was not preferred due to limbus and LASIK flap involvement. Instead, ProKera^®^ was chosen to cover the limbal area, promoting stem cell proliferation and aiding in wound healing. With the application of AM, deep resection of the tumor over the limbus could be performed confidently, which reduced the risk of recurrence. 

Previous studies used fibrin glue-assisted sutureless AM transplant in patients with corneal limbal dermoid excision, and the results showed rapid corneal re-epithelialization and reduced postoperative pain and scarring [9,10]. Asoklis et al. [11] used AM following total lesion-free conjunctival and limbal tumor excision, including 2 squamous cell carcinomas, 2 papillomas, and 5 nevi. After a mean 38-month follow-up, 8 cases (88.9%) had complete healing, and 2 cases (22.2%) had partial limbal stem cell deficiency. Only 1 case of papilloma (11.1%) had recurrence after 3-year follow-up. Others applied sutured AM following excision of conjunctival and limbal tumors, with similar results [12]. In particular, our case 2 was PAM with atypia overlying the cornea and LASIK flap. The complete resection of tumors might damage the LASIK flap and increase the risk of epithelial ingrowth. The literature revealed the effect of an AM pressure patch inhibited epithelial ingrowth in cases of LASIK flap trauma. Kwon et al. applied the AM over the cornea and sutured tightly to the episclera, and no recurrent epithelial ingrowth occurred after a 5-month follow-up [13,14]. Meskin et al. [15] sutured the central LASIK flap in cases of epithelial ingrowth following perforated cornea injury. No recurrence was noted after 1-year follow-up, but 1.25 diopter astigmatism remained. The application of sutureless AM promotes wound healing and decreases the risk of recurrence and postoperative astigmatism.

AM can be used in cases of pterygium excision [16]. Excision with adjunctive mitomycin C or 5-fluorouracil could reduce the recurrent rate but increase the risk of postoperative scleral melting. Thus, the conventional strategy to prevent recurrence is pterygium excision with conjunctival autograft or alternative with amniotic membrane graft. The AM graft could be secured with sutures or fibrin glue. However, sutures may cause stitch reaction and elevate the risk of stitch abscess, granulation formation and infection [17]. Sutureless AM graft with fibrin glue has the potential to drop due to gravity or during blinking. With the sutureless AM to cover the wound, it promotes cornea/limbus/conjunctiva healing and reduces the anti-neovascularization, scarring and stitch reaction. The use of ProKera^®^ following pterygium excision saves time, as well as preserves most normal conjunctiva tissue with less bare sclera, which reduces the risk of scleral melting.

Multiple studies have mentioned the importance of early AM transplant within 3 to 7 days in cases of Stevens–Johnson syndrome (SJS) [18,19]. AM grafting over the entire bulbar conjunctiva, palpebral conjunctiva, fornices and lid margins has been suggested to prevent the risk of irreversible scarring [20,21]. However, in many SJS cases, the systemic condition is devastating, such as respiratory failure, which makes patients not suitable for invasive surgical intervention. ProKera^®^ offers the advantage of an easily performed technique, making it readily available for non-ophthalmic physicians, particularly in the bedside settings of intensive care units or emergency rooms [22]. We demonstrated one case of acute SJS (case 6) with complete remission and 2 cases of chronic SJS (cases 4 and 5) with chronic inflammation sequela. Patients with the chronic phase of SJS might have symblepharon, lid margin keratinization, meibomian gland injury, punctal occlusion, limbal stem cell deficiency, cornea conjunctivalization or neovascularization [18]. The current hypotheses of chronic SJS include persistent inflammation of the ocular surface and an increase in inflammatory cellular infiltration and cytokines [23]. Although there are many management strategies for chronic SJS, including autologous mucous membrane grafting for lid margin keratinization, punctal occlusion for meibomian gland dysfunction, and limbal stem cell transplantation for limbal stem cell deficiency, the interventions become invasive and visual recovery is more difficult [24]. Consistent with the literature, early AM is essential for SJS cases. ProKera^®^ treatment provides an easy method for patients with critical systemic conditions and are unable to undergo surgical interventions.

This study has limitations inherent in a retrospective study and small cases. In most cases, complete healing could be achieved in short-term periods, while long-term follow-up is needed, especially in cases with corneal-limbal-conjunctival tumors, to evaluate the recurrence rate. However, our study’s strength lies in the comprehensive documentation of external photos in patients with acute/chronic SJS, showcasing varying severities within the same patient. Additionally, we expanded the clinical application of ProKera^®^ by employing it in various diseases. Notably, case 2 was the first documented case of ProKera^®^ use in a patient with recurrent pigmented conjunctival lesions involving both the cornea and a prior LASIK flap. Additionally, ProKera^®^ costs patients TWD 30,000 per piece, compared to TWD 4000 of sutured fresh AM, which was covered by national health insurance in Taiwan. The high price might limit the generalization. However, a recent study compared the expense of ProKera^®^ and suture AM transplant and they concluded ProKera^®^ is more cost-effective. They found the cost of ProKera^®^ was CAD 699.00, whereas AMT had a total cost of CAD 1561.52. For each percentage healed surface area, ProKera^®^ priced at CAD 11.85, compared to CAD 21.39 for AMT [25]. In addition, ProKera^®^ takes advantage of implementations in outpatients. Early intervention with ProKera^®^ in patients who are not suitable candidates for invasive surgeries reduces the cost of treating the long complication of chronic SJS. We conclude that self-retained cryopreserved AM is effective following deep resection of conjunctival-limbal-corneal lesions or in patients with Stevens–Johnson syndrome. An easily applicable ProKera^®^ provides a treatment option for non-ophthalmic physicians to intervene earlier in cases of SJS.

## Figures and Tables

**Figure 1 jcm-12-06222-f001:**
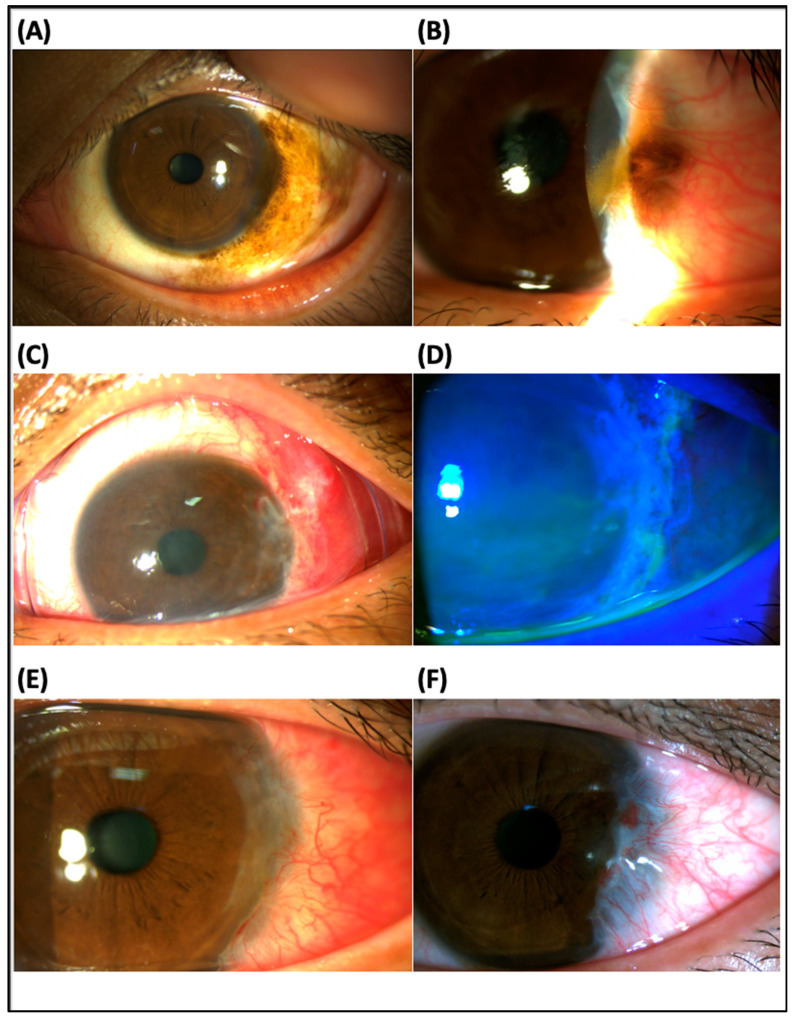
Slit lamp examination in a patient with primary acquired melanosis (PAM) with atypia (case 2): (**A**) The initial findings showed a pigmented conjunctival lesion in the right eye. The pathology revealed PAM with mild atypia. (**B**) A recurrent pigmented conjunctival lesion with corneal and laser-assisted in situ keratomileusis (LASIK) flap invasion at the same location occurred 19 months post-excision. (**C**,**D**) Conjunctival, limbal and corneal defects healed 3 days after tumor excision, keratectomy for the conjunctival, limbal and corneal lesion, and ProKera^®^ insertion. (**E**) There was no epithelial downgrowth or other flap complication 1 month after operation. (**F**) Superficial peripheral vascularization was found 7 months post-operation. The LASIK flap was intact, and there was no tumor recurrence during an 11-month follow-up. The new pathology was PAM with mild atypia.

**Figure 2 jcm-12-06222-f002:**
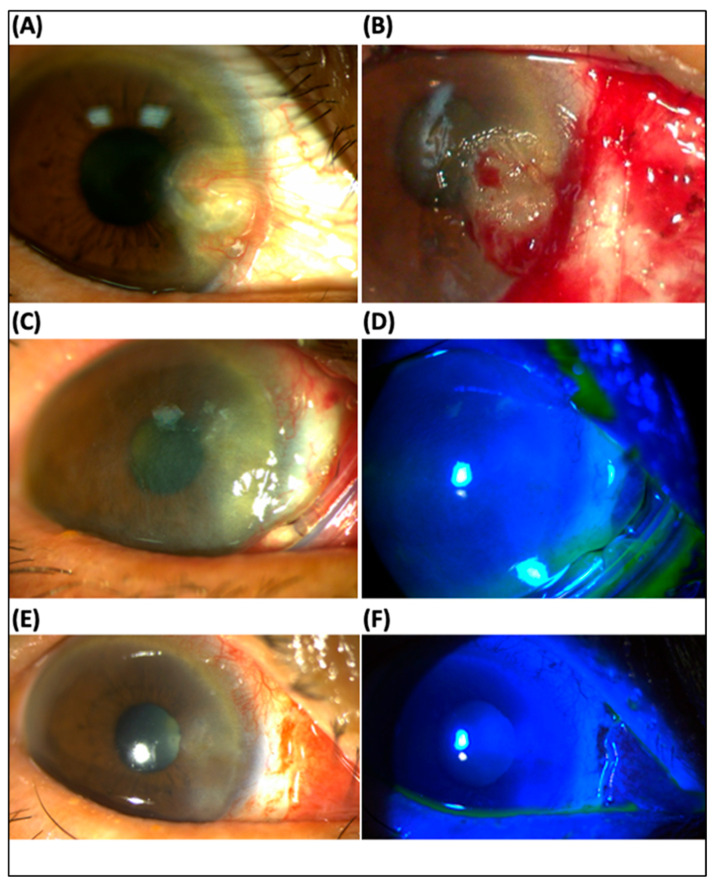
Slit lamp examination in a patient with pterygium excision (case 3): (**A**) The initial findings showed a pterygium with corneal involvement in the right eye, which caused high astigmatism (best corrected visual acuity, BCVA 6/30 in Snellen chart, anterior cornea curvature cylinder −7.0 D). (**B**) Post-keratectomy, pterygium excision with bare sclera 2 mm and insertion of ProKera^®^ (**C**,**D**) Complete healed corneal defects with mild haziness 3 days postoperatively. (**E**,**F**) During post-operative 6-day follow-up, ProKera^®^ was removed. There was complete healing of the cornea and conjunctiva. BCVA improved to 6/6.7 in the Snellen chart, and astigmatism resolved (+2.0, −0.5 × 144).

**Figure 3 jcm-12-06222-f003:**
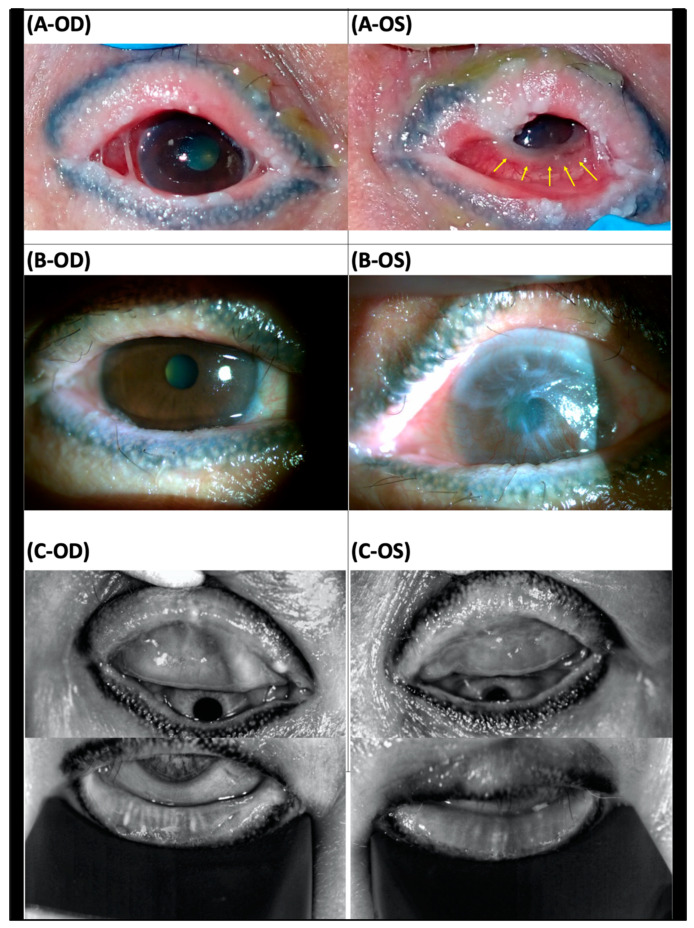
Slit lamp examination and meibomian photographs in a patient with chronic Stevens–Johnson syndrome (SJS) (case 5): (**A**) Two months after onset of SJS, the patient first came to our clinic with the presentation of prominent symblepharon, discharge, severe congested conjunctiva in both eyes, and severe inflammation over the limbus area in the left eye (arrow). (**B**) Following 4 months of SJS onset and 2 months of ProKera^®^ use, corneal conjunctivalization and vascularization in the left eye developed. A clear cornea was noted in the right eye. (**C**) Meibomian gland (MG) photographs revealed diffuse MG shortening in the lower part and total loss of MG in the upper part. The left side was more severe than the right side. Vision improved to 6/6 in the right eye but was poor, as 3/60 in the left eye, in a 7-month follow-up.

**Figure 4 jcm-12-06222-f004:**
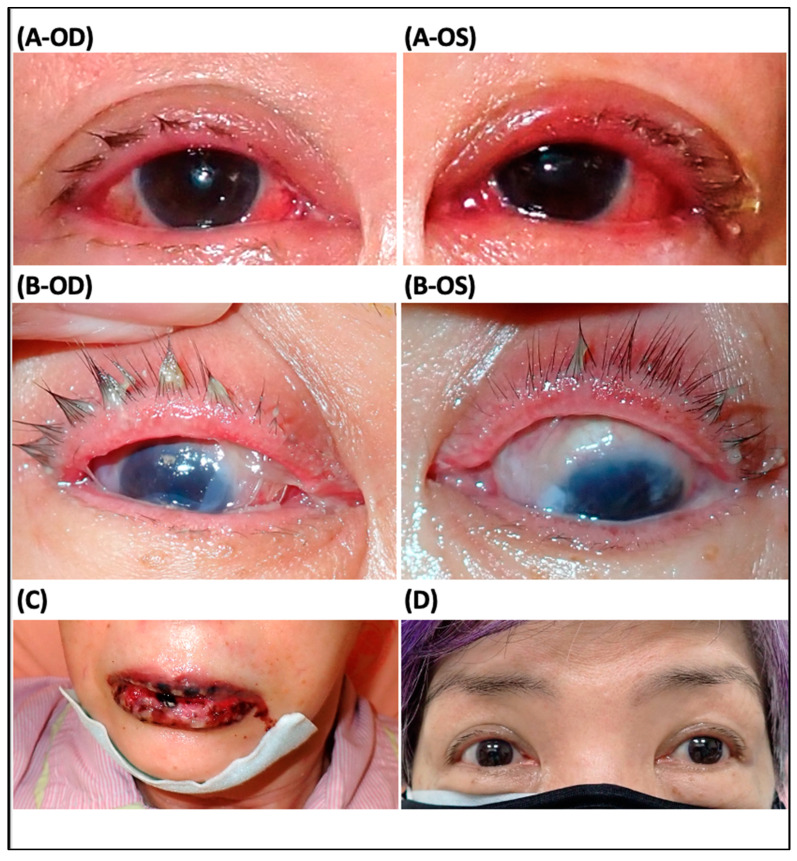
Slit lamp examination in a patient with acute Stevens–Johnson syndrome (SJS) (case 6): (**A**) Eyelid and conjunctiva swelling in both eyes were noted 7 days after acute SJS onset. (**B**,**C**) Following 9 days of SJS onset, there was prominent discharge and pseudomembrane formation in both eyes and severe oral mucosal erosions. ProKera-classic^®^ was inserted into both eyes. (**D**) Following 3 months of SJS onset and ProKera^®^ use, there was no conjunctival inflammation, no corneal defect, no symblepharon formation, or meibomian gland disorder. Vision achieved 6/6 in both eyes and was maintained at a 6-month follow-up.

**Table 1 jcm-12-06222-t001:** Clinical demographics and treatment outcomes in patients with self-retained cryopreserved sutureless amniotic membrane (ProKera^®^) use.

Case	Age/Gender	Diagnosis	Clinical Presentation	Treatment, ProKera^®^ (Total Use of Number)	Total Follow Period (Months), Outcome	BCVA (Initial)	BCVA (Final)
1	25/F	Recurrent conjunctival nevus	Diffuse pigmented conjunctival lesion with corneal invasion, OS	excision + keratectomy + ProKera^®^-slim (1)	12 months, No recurrence	6/6.7	6/6.7
2	36/M	Recurrent PAM with atypia	Recurrent pigmented conjunctival lesion with corneal and LASIK flap invasion, OS	excision + keratectomy + ProKera^®^-slim (1)	11 months, No recurrence Intact LASIK flap	6/6	6/6
3	69/F	Pterygium	Pterygium with marked corneal involvement, OD	excision + keratectomy + ProKera^®^-slim (1)	1 month, Astigmatism resolved with vision improvement	6/30	6/6.7
4	74/M	Chronic SJS (ProKera^®^ was used after 2 months of SJS onset)	Severe symblepharon, cornea and eyelid epithelial defect, OU	lubricants, topical steroids, symblepharon release, ProKera^®^-classic (1 OD + 1 OS)	10 months, Symblepharon and inflammation resolved, meibomian gland dysfunction	OD 6/30 OS 6/15	OD 6/15 OS 6/15
5	69/F	Chronic SJS (ProKera^®^ was used after 2 months of SJS onset)	Severe symblepharon, eyelid/cornea/conjunctiva ulceration, OU; Severe keratinization in the limbus, OS	lubricants, topical steroids, symblepharon release, ProKera^®^-classic (2 OD + 7 OS)	7 months, Symblepharon and inflammation resolved, meibomian gland dysfunction in both eyes, corneal conjunctivalization with neovascular formation in the left eye	OD 6/12 OS CF	OD 6/6 OS 3/60
6	68/F	Acute SJS (ProKera^®^ was used after 9 days of SJS onset)	Eyelid/cornea/conjunctiva inflammation, cornea and eyelid epithelial defect, OU	lubricants, topical steroids, ProKera^®^-classic (1 OD + 1 OS)	6 months, Complete remission of symptoms and signs of eyelid/conjunctiva/cornea	OD 6/7.5 OS 6/10	OD 6/6 OS 6/6

PAM: primary acquired melanosis; LASIK: laser-assisted in situ keratomileusis; SJS: Stevens–Johnson syndrome; BCVA: best corrected visual acuity; CF: counting finger; OD: right eye; OS: left eye; OU: both eyes.

## Data Availability

Not applicable.

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
