# Peer review of "Self-Retained, Sutureless Amniotic Membrane Transplantation for the Management of Ocular Surface Diseases"

_jcm, 2023, doi:10.3390/jcm12196222_

Round 1

Reviewer 1 Report

The article is interesting, and provides treatment for some pathologies in which ProKera® can be useful.

1. Correct in abstract: Jan2022 and May2023

2. Perhaps proposing in the abstract its use not ophthalmic physicians is not appropriate. For two reasons: cost of the product and risk of the surgery itself, such as Steven-Johnson syndrome.

3. Introduction: Add that: it has lower concentrations of proteins/growth factors.

Cryopreserved AM retains similar histological properties to fresh AM, however, it has lower concentrations of proteins/growth factors.

4.  Case 6. Specify drugs used? lubricants and anti-inflammatory eyedrops

5. In discussion, it is not clear why some patients are operated on with Prokera and not with another technique. If it is due to hypofunction or total loss of stem cells?. The usefulness of AM is in the face of stem cell hypofunction or total loss of stem cells. It is an idea little transmitted at work.

6. Improve the expression of this paragraph: In cases with recurrent conjunctival tumors with corneal involvement, residual conjunctival lesions in the limbus are the main cause of recurrence which we dare not remove too deep in the limbus in the initial operation. Deep resection could reduce tumor recurrence. However, limbus contains stem cells, damage of limbus causes the risk of wound healing and proliferation. With the application of AM, deep resection of tumor

7. Better to express it: (Author, year) who publishes these results: in Literature revealed the effect of an AM. 

8.  However, it is difficult to calculate the cost of time and human resources . Remove this phrase or provide information supported by the bibliography.

9. The most important thing about this work is the good results obtained in patients with Steven-Johnson syndrome. The persistence of AM is very limited in days, the previous surgical technique and the postoperative treatment are very important and are listed in a vague way, at least in these cases more precision would be desirable.

10. Publications on ProKera are beginning to increase. It would be interesting to highlight the strengths and weaknesses of this work, the latter being somewhat more developed.

Reviewer 2 Report

This is an article entitled “Self-retained, sutureless amniotic membrane transplantation for the management of ocular surface diseases (jcm-2596024)” which evaluates the efficacy of ProKera® in ocular surface diseases.

Abstract

-          Please give the ranges of all data.

Introduction

-          Good.

Methods

-          Please admit the postoperative treatment regimens as well.

Results

-          For Case 1 as it was a recurent lesion why did not you treat it as a malign lesion to perform adjunctive double-freeze thaw cryotherapy to the conjunctival margins after excision?

-          For Case 2 the one with PAM with mild atypia what treatment was performed? Did you add any additional drops etc such as INF-alpha2b or 5-FU? Please declare. If you had any suspicions earlier why did not you perform double-freeze thaw cryotherapy to the margins of the conjunctiva after lesion excision?

Discussion

-          What is the advantage of ProKera to cryopreserved conventional amniotic membrane? As it is a very expensive treatment please declare these advantages if any.

-          As you know the use of amniotic membrane for all the cases you mentioned are well known. So you should mention the differences if any.

References

-          There should be more differences that consider the efficacy of amniotic membran efor the cases you presented.

Tables

-          Okay.

Figures

-          Ok.

English is okay.

Round 2

Reviewer 1 Report

Just fix this minor fix: References Remove some reference like this: 1. JWD, Skin transplantation with a review of 550 cases at the John's Hopkind Hospital. Johms Hopkins Med J 15:307. 1910. And change to some more current reference focused on your work: Lacorzana J, Campos A, Brocal-Sánchez M, et al. Visual Acuity and Number of Amniotic Membrane Layers as Indicators of Efficacy in Amniotic Membrane Transplantation for Corneal Ulcers: A Multicenter Study. J Clin Med. 2021 Jul 22;10(15):3234. doi:10.3390/

Author Response

Please see the attachment》

Reviewer 2 Report

All asked changes are made.

Author Response

Thank you for taking the time to review our manuscript and offer valuable feedback and comments. On behalf of all the authors of this manuscript (manuscript number jcm-2596024), I submit our responses to the reviewers’ comments and suggestions.  

Comment 1: All asked changes are made.

Response 1: Thank you for taking the time to review our manuscript.
